# Exploring the causes underlying the latitudinal variation in range sizes: Evidence for Rapoport's rule in spiny lizards (genus *Sceloporus*)

**Kevin López-Reyes**[1,2,3]*, **Carlos Yáñez-Arenas**[2]*, **Fabricio Villalobos**[3]*

**1** Posgrado en Ciencias Biológicas, Universidad Nacional Autónoma de México (UNAM), Mérida, México, **2** Laboratorio de Ecología Geográfica, Unidad Académica Sisal, Facultad de Ciencias, Unidad de Conservación de la Biodiversidad, Parque Científico y Tecnológico de Yucatán, Universidad Nacional Autónoma de México, Mérida, Yucatán, México, **3** Laboratorio de Macroecología Evolutiva, Red de Biología Evolutiva, Instituto de Ecología A.C.–INECOL, Xalapa, Veracruz, México

* lopezreyes.ka@gmail.com (KLR); lichoso@gmail.com (CYA); fabricio.villalobos@gmail.com (FV)

**Data Availability Statement:** The data underlying the results presented in the study are available from GBIF (www.gbif.org; https://doi.org/10.15468/dl.5234bm) and WorldClim (https://www.worldclim.org/data/worldclim21.html).

## Abstract

Species' range size is a fundamental unit of analysis in biodiversity research, given its association with extinction risk and species richness. One of its most notable patterns is its positive relationship with latitude, which has been considered an ecogeographical rule called Rapoport's rule. Despite this rule being confirmed for various taxonomic groups, its validity has been widely discussed and several taxa still lack a formal assessment. Different hypotheses have been proposed to explain their potential mechanisms, with those related to temperature and elevational being the most supported thus far. In this study, we employed two level of analyses (cross-species and assemblage) to investigate the validity of Rapoport's rule in spiny lizards (genus *Sceloporus*). Additionally, we evaluated four environmental-related hypotheses (minimum temperature, temperature variability, temperature stability since the last glacial maximum, and elevation) posed to explain such pattern, contrasting our results to those patterns expected under a null model of range position. Our results provided support for Rapoport's rule at both levels of analyses, contrasting with null expectations. Consistently, minimum temperature and elevation were the most relevant variables explaining the spatial variation in range size. At the cross-species level, our null simulations revealed that both variables deviated significantly from random expectations. Conversely, at the assemblage level, none of the variables were statistically different from the expected relationships. We discussed the implication of our findings in relation to the ecology and evolution of spiny lizards.

## 1. Introduction

In ecology and biogeography, the concept of "species distributional area" refers to the geographical regions of the planet where individuals of a specific species have been observed to

**Funding:** KLR received support from the Consejo Nacional de Humanidades Ciencias y Tecnologías (CONAHCYT; CVU: 1010702). The funders had no role in study design, data collection and analysis, decision to publish, or preparation of the manuscript.

**Competing interests:** The authors have declared that no competing interests exist.

occur. These areas encompass the ecological conditions that allow the species to have non-ephemeral interactions with the environment [1, 2]. Previous studies have confirmed that the size of species' distributional areas, also known as range size, is inversely related to species richness [3] and with the risk of extinction [4]. Consequently, understanding the spatial variation of range size can provide valuable insights into global patterns of biodiversity and inform conservation actions. One of the best-known macroecological patterns describing this variation is the positive relationship between range size and latitude, also known as Rapoport's rule [5]. Although Rapoport's rule has been supported for various taxonomic groups [6–8], it's validity has received strong criticism. For instance, the idea that this rule might be only an effect observed primarily in regions of the Northern Hemisphere, especially in the Americas [9–11]. Furthermore, despite the great interest in this rule, several species-rich groups still lack a proper assessment of their range size latitudinal patterns and their potential causes. Reptiles are an example of such groups, as they have received less attention compared to other extensively studied groups of terrestrial vertebrates, such as birds [8, 12, 13], mammals [6, 14], and amphibians [15]. Among reptiles, lizards have received less attention in the study of macroecological patterns and their potential mechanisms [16, 17], which poses a challenge to the generalization of Rapoport's rule.

A crucial step towards addressing this knowledge gap consists in evaluating the existence of Rapoport's rule in such groups as well as understanding the mechanisms driving the spatial variation in their range sizes. Indeed, several hypotheses have been proposed to explain Rapoport's rule with those related to temperature (current and past) and elevation being the most supported, mainly due to their influence on species' physiological tolerance and dispersal abilities [15, 18–20]. For instance, the "climatic variability hypothesis" [5] suggests that species present at temperate latitudes tend to be thermal generalists due to the high temperature variability experienced in such areas, which in turn allows them to have larger range sizes compared to their tropical counterparts. Alternatively, Pither [21] suggested the "climatic extremes hypothesis" in which, thermal limits are what restrict range sizes. According to this hypothesis, species able to tolerate more extreme temperature limits, namely cold ones, will exhibit larger range sizes. Another hypothesis posed to explain Rapoport's rule centers on historical climatic stability [22]. Regions with stable climates over extended periods of time, such as the tropical regions, facilitate the persistence of species with small ranges. In contrast, areas characterized by greater climatic variation over time, such as those in temperate latitudes, primarily support species with larger range sizes. This is because species with smaller range sizes are more susceptible to extinction processes such as those exerted by climatic instability [22–24]. One additional hypothesis to explain Rapoport's rule relates to elevation [15, 25]. The ability of species to move across elevation gradients is influenced by the climatic variability they encounter [26]. In the tropics, there is greater temperature variation across elevation gradients compared to temperate regions. Therefore, tropical mountains act as more restrictive physiological barriers, causing tropical species to face greater challenges in expanding their ranges across these gradients. Consequently, tropical mountain species tend to have smaller range sizes compared to those in temperate regions [25].

In addition to these environmental hypotheses, other models have emerged in an attempt to explain the spatial variation in range sizes. For instance, the Mid-Domain Effect (MDE) provides a "non-biological" explanation for species distribution patterns, attributing them to the geometric constraints imposed by the shape of the geographic domain [27]. According to this model, species' ranges are randomly arranged within a domain, where regions with greater availability of area (usually the center of the domain) tend to have greater overlap in species ranges (hence higher species richness) as well as larger range sizes [9, 27]. This scenario can be recreated using null models that randomize the geographic position of the species' ranges

while maintaining their size [28]. This allows breaking the relationship between the environment and the position and size of the geographic range, making them useful for comparing the environmental variation experienced by species in their original positions with null observations where theoretically the environment has no effect on range size [6, 29, 30]. Consequently, it is crucial to include null models when assessing latitudinal gradients in range sizes and their relationship with potential determinants.

Compared to other vertebrate groups, and especially in North America, there is a noticeable knowledge gap regarding the range size pattern in lizards. To our knowledge, no formal analysis has been done to evaluate the existence of Rapoport's rule nor the proposed hypotheses to explain it in North American lizards. Spiny lizards (genus *Sceloporus*) represent an ideal group to fill this gap. This genus encompasses ~116 species distributed across a broad latitudinal extent and thus thermal gradient as well as ample topographic variation [31, 32]. Likewise, given their ectothermic physiology, these species are expected to be sensitive to changes in environmental temperature, which could affect their activity patterns [33], locomotion [34], reproductive strategies [35, 36], and consequently, their geographic distributions. For this genus, it has been observed that tropical species have narrower thermal tolerance and elevation ranges compared to their temperate counterparts [35]. Additionally, temperate species within this genus may maintain body temperature more effectively than tropical species, which affects their mobility and dispersal capacity [37]. However, despite extensive research on phylogenetic [38, 39], ecological [40–42] and evolutionary [31, 35, 43–46] aspects within this genus, its main macroecological patterns remain unevaluated.

In this study, we investigated the existence of Rapoport's rule in the genus *Sceloporus* and assessed the importance of four environmental hypotheses posed to explain this rule (climatic variability, climatic extremes, historical climatic stability, and elevation; see Table 1) within this group. We hypothesize that the geographic range size of *Sceloporus* will increase with latitude due to the influence of environmental factors such as temperature and elevation. To evaluate this, we performed spatial and phylogenetic regressions and contrasted our results with null models to determine if the observed patterns can be recreated assuming no effect of environmental gradients at two levels of analysis (cross-species and assemblage). This research represents the first formal assessment of Rapoport's rule for a lizard group in North America, as well as its potential determinants, and provides valuable insights into geographic and ecological knowledge about spiny lizards of the genus *Sceloporus*.

## 2. Materials and methods

All data cleaning procedures, model construction, and statistical analyses mentioned in the subsequent sections were conducted using the R software version 4.2.2 [49] and QGIS version 3.32 [50].

### 2.1 Geographic data

**2.1.1 Occurrence records.** Presence data, namely occurrence records, of *Sceloporus* species were obtained through Global Biodiversity Information Facility [51] and Naturalista [52]. The information from both repositories was integrated into a single species-level dataset, where taxonomic harmonization was performed following Uetz *et al*. [32] as the taxonomic authority. Then, records without geographic coordinates and duplicates were removed using the function "clean_dup" from the "ntbox" package [53]. Due to the inherent biases associated with sampling efforts and geographical inaccuracies, we performed an additional cleaning process. First, to reduce spatial autocorrelation, we utilized the "gridSample" function from the "dismo" package [54]. This function allowed us to retain a single presence record per cell,

**Table 1. Environmental hypotheses posed to explain the Rapoport's rule evaluated in this study.**

| Hypothesis | Prediction | Explanation | Variable | Source |
|---|---|---|---|---|
| Climatic variability [5] | + | Species found in temperate latitudes tend to be thermal generalists, which allows them to have larger range sizes. | Temperature annual range | WorldClim [47] |
| Climatic extremes [21] | - | Species that are able to tolerate more extreme temperature limits, namely colder ones, will exhibit broader distribution areas. | Minimum temperature of the coldest month | WorldClim [47] |
| Historical climatic stability [22, 23] | + | Greater climatic variation over time, predominantly supports species with larger range sizes. As species with smaller ranges are more susceptible to extinction. | Climate change velocity | Sandel *et al.* [24] |
| Elevation [25] | - | Tropical mountains pose greater physiological barriers for species to expand their range sizes. | Digital elevation model | Shuttle Radar Topography Mission [48] |

based on a 10 arc-minute raster resolution, which corresponds to approximately 340 square kilometers. Then, to eliminate geographically atypical data, we employed the "cc_outl" function from the "CoordinateCleaner" package [55]. In this process, we initially calculated the distances to the nearest neighbors for each presence record. If the distance exceeded two times the interquartile range of the average distance of all observations, the record was classified as a geographically atypical observation and removed from the analysis. The final dataset of occurrence records comprised a total of 24,336 records, where *S. esperanzae*, *S. grandaevus*, and *S. macdougalli* had the lowest number (each with three unique points), while *S. undulatus* exhibited the highest number with 3,836 unique records (S1 Table).

**2.1.2 Geographic ranges.** Although several previous studies have evaluated Rapoport's rule in one dimension using latitudinal extent as the observed variable [5], it has become evident that this unidimensional approach fails to capture the multidimensional nature of geographical ranges [56, 57]. As such, geographical data should ideally be analyzed in the same dimensional space in which they occur [58]. Accordingly, most recent studies evaluating Rapoport's rule now consider areal extent or size instead of simply latitudinal extent to evaluate the rule and its underlying causes [14, 20]. We followed this approach and estimated range size of each species as its total occupied area (in km$^2$; see section 2.4 below). To do so, we generated geographic ranges (i.e., extent of occurrence, [1]) for each *Sceloporus* species based on their compiled and cleaned occurrences records using three different methods for range construction to ensure that range size estimates were not method dependent.

We utilized one method based on a correlative approach between occurrences and climate (species distribution modeling), as well as two polygon-based estimations: **1) convex hulls:** defined as the minimum convex set that contains a finite series of points, connecting the outermost points, where internal angles are less than 180 degrees [59] and **2) alpha-hulls:** which allow the construction of non-convex sets by adjusting the shape of convex polygons based on an alpha parameter, resulting in shapes that may include holes or empty spaces within [60].

Given that the polygon construction is based on Voronoi triangulation, for the alpha and convex hulls we only considered species with a minimum of three distinct occurrence records after data depuration. To construct the convex-hulls, we used the "rangemap_hull" function from the "rangemap" package [61]. Alpha-hulls were estimated using the "getDynamicAlphaHull" function from the "rangeBuilder" package [62], which starts with an initial alpha value and gradually increases it until two conditions are met: 1) the desired percentage of points to be included within the polygon (in our case, 100%) and 2) the maximum number of disjoint distributions allowed (since this study did not consider disjoint distributions, only one alpha polygon was generated per species). Both methods used a buffer of 50 km, which represents the observed maximum distance of population separation within genus *Sceloporus* [63].

To construct the species distribution models we employed the maximum entropy algorithm Maxent [64]. This analysis considered only those species with a minimum of seven occurrence records after data cleaning. The occurrence records were then divided into two subsets: training and validation. For species with less than 25 occurrence records, a random partitioning was performed using 60% of the records for training and 40% for validation. The remaining species' records were partitioned using the "get.checkerboard1" function from the "ENMeval" package [65]. This function employs a checkerboard pattern to divide the observations, using the cells of an input raster as the basis. In our case, the raster had a resolution of ~340 km$^2$. After this, we collected the 19 bioclimatic variables from WorldClim version 2.1 (www.worldclim.org), which represent annual temperature and precipitation patterns between the years 1970 and 2000 [47]. Additionally, a digital elevation model derived from the Shuttle Radar Topography Mission (SRTM; https://earthexplorer.usgs.gov) was included in the analysis [48]. The variables from both repositories were obtained at a ~340 km$^2$ resolution. Initially, bioclimatic variables 8, 9, 18, and 19 were removed because, as observed in previous studies, we noticed artifacts in its construction [66]. Afterwards, a variance inflation factor analysis was conducted to reduce multicollinearity among variables using the "vifstep" function from the "usdm" package [67]. Based on this procedure, the variables used for modeling were the digital elevation model and the following bioclimatic variables: 2 (mean diurnal range), 7 (temperature annual range), 10 (mean temperature of warmest quarter), 13 (precipitation of wettest month), 14 (precipitation of driest month), and 15 (precipitation seasonality). These variables were cropped to a calibration region for each species, based on the convex-hulls. Then, for each species, optimal parameters for modeling were obtained through a series of candidate models, which were constructed and evaluated using the "kuenm_cal_swd" function from the "kuenm" package [68]. Based on this function, we generated the following combinations for different types of response curves: "linear", "linear and quadratic", "linear, quadratic and product", "linear, quadratic, product and threshold" and "linear, quadratic, product, threshold and hinge". We also tried different regularization multipliers (1, 2, 3, 4). After, the selection of the best models was done using three hierarchical criteria: 1) statistical significance: models that performed better than expected by chance, i.e., the average area under the curve greater than 1, using partial receiver operating characteristic analysis [69], 2) predictive ability: we selected models that correctly anticipated at least 90% of the occurrence records, resulting in a 10% omission rate [64] and 3) complexity: using the small-sample corrected Akaike Information Criterion, we selected models with a difference of fewer than two units of the best model [70]. Finally, using the "kuenm_mod_swd" function, the final model was transferred to a region representing a hypothesis of historical accessibility for each species, referred to as "M" sensu Soberón and Peterson [71], constructed with terrestrial ecoregions [72]. To perform this, we included into the "M" those ecoregions that had at least one occurrence record. Subsequently, the continuous models generated through Maxent modeling were transformed into binary representations of distribution areas using the "rangemap_enm" function from the "rangemap" package [61]. The binarization threshold to obtain the final range map of each species was determined as the minimum value that encompassed the 90% of environmentally nearest records.

## 2.2 Environmental predictors of range size

To evaluate the environmental hypotheses posed to explain Rapoport's rule, we used a set of predictor variables. The climatic variability hypothesis was assessed using the temperature annual range, whereas the climatic extremes hypothesis was evaluated using the minimum temperature of the coldest month. Both variables were obtained from WorldClim version 2.1

(www.worldclim.org; [47]). The hypothesis of elevation was evaluated using the previously mentioned digital elevation model [48]. To represent historical climatic stability, we used the climate change velocity since the last glacial maximum (~21,000 years ago), obtained from Sandel *et al.* [24]. This variable represents a measure of the speed at which climatic conditions change in a particular region [73], obtained by dividing a temporal gradient (in this case, the differences in annual mean temperature between the current and the last glacial maximum) by a spatial gradient (interpreted as the rate of temperature change relative to the distance between adjacent cells), resulting in a variable expressed in units of distance per time. All the aforementioned variables were transformed to an "Albers Equal Area" projection using the "projectRaster" function from the "raster" package [74] and obtained at a spatial resolution of ~340 km$^2$, except for the climate change velocity, which was obtained at a resolution of 2.5 arc-minutes (~21.6 km$^2$). However, in order to match the presence-absence matrix described in section 2.5 below, these variables were resampled to a resolution of 50 km × 50 km using the "resample" function from the "raster" package [74].

## 2.3 Phylogenetic relationships

To consider the phylogenetic relationships among species in our analyses, we obtained two previously published phylogenies, which are the most recent and comprehensive phylogenies including the genus *Sceloporus*. The first phylogeny was obtained from Leaché *et al.* [38], which included 129 phrynosomatid species, of which 86 belong to the genus *Sceloporus* [32]. This phylogeny also included six subspecies of *Sceloporus* (*S. magister cephaloflavus*, *S. magister magister*, *S. magister uniformis*, *S. formosus scitulus*, *S. grammicus microlepidotus* and *S. megalepidurus pictus*), and two species without current taxonomic validity at the moment of this study: *S. vandenburgianus*, currently considered as a subspecies of *S. graciosus* [75] and *S. lineolateralis*, at present considered as a subspecies of *S. jarrovi* [76, 77]. These latter species, as well as the aforementioned subspecies, were removed from the phylogeny to match with the most recent taxonomic update [32]. More specifically, we used the maximum clade credibility tree provided by Leaché *et al.* [38], which was produced in BEAST based on concatenation and coalescent-based species tree inference using 585 nuclear loci (541 ultraconserved elements and 44 protein-coding genes) and calibrated with an uncorrelated log-normal relaxed clock with a single calibration.

The second phylogeny was obtained from the species-level megaphylogeny for the entire order Squamata produced by Tonini *et al.* [39]. These authors generated fully-sampled phylogenies for the whole order Squamata, encompassing 9,574 species of which 97 belongs to *Sceloporus*. First, they created a maximum-likelihood (ML) topology using ExaML/RaxML for 5,415 species based on a molecular supermatrix with sequence data for 17 genes (7 mitochondrial and 10 nuclear). Then, using MrBayes 3.2 [78] they dated several subclades under a relaxed-clock model with node-age calibrations. Finally, they included the remaining species (those lacking molecular data) to the ML topology by randomly assigning them within their genus or higher-level clade. From their resulting posterior distribution of 10,000 trees, we generated a maximum clade credibility tree based on the median branch lengths using the software TreeAnnotator v1.10.4 [79].

## 2.4 Cross-species and assemblage level approaches

For the following analyses, we considered three datasets: 1) all species, 2) those contained in Leaché *et al.* [38] and 3) the ones present in Tonini *et al.* [39]. We include only those species for which we were able to estimate their geographic ranges. Thus, of the 116 species reported for the genus [32], in the subset of "all species", we had 103 species, while in subsets of Leaché

*et al.* [38] and Tonini *et al.* [39], there were 81 and 92 species, respectively. First, we constructed a presence-absence matrix, where columns represent species and rows sites with binary entries representing presence (1) or absence (0), using a grid-cell system with resolution of 50 × 50 km and an "Albers Equal Area" projection using the "lets.presab" function from the "letsR" package [80].

To robustly evaluate Rapoport's rule, we considered two complementary levels of analyses considering the two dimensions of the pattern: the cross-species and assemblage levels [81]. In the cross-species level, species are considered as units of observation such that the variables (response and predictors) of interest are measured for each individual species, whereas in the assemblage level the units of observation are the assemblages of species present in each site (i.e., grid-cell) thus the variables are summarized within them. At the cross-species level, we obtained the range size in km$^2$ (response variable) and latitudinal midpoint (explanatory variable) for each species from the presence-absence matrix. These attributes were calculated using the "lets.rangesize" and "lets.midpoint" functions respectively, both implemented in the "letsR" package [80]. Given the large variation in magnitude for the range sizes, this variable was log-transformed. To ensure that the estimation of these properties was not dependent on the method used to construct the geographic ranges, we compared their range sizes and latitudinal midpoints using Spearman's rank correlation coefficient. We found that three methods produced highly correlated range sizes and midpoints (S1 Fig), so we only used the method that demonstrated more conservative estimates of range size (alpha-hulls) for downstream analyses. At the assemblage level, our response variable was defined as the median of the log-transformed range size for all the species present in a given cell. To perform this calculation, we employed the "lets.maplizer" function [80]. Meanwhile, our explanatory variable was the latitudinal centroid corresponding to each cell in the presence-absence matrix.

To evaluate the environmental hypotheses, we used the same response variable as in the evaluation of Rapoport's rule, log-range size of species and median log-range size of assemblages for the cross-species and assemblage level analyses, respectively. For the explanatory variables, we considered two distinct approaches, depending on the level of analysis. For the cross-species level, we considered the mean climatic values within the geographic range of each species. For the assemblage level, we focused on the values of each climatic variable within each grid-cell.

## 2.5 Statistical analysis

All predictor and response variables were standardized using the "scale" function from R base. The mean was set to 0 and the variance was set to 1. We also checked for multicollinearity problems using the "vif" function from the "usdm" package [67].

**2.5.1 Phylogenetic signal of species attributes.** To investigate if the considered species-level attributes (e.g., range size, latitudinal position, mean climatic conditions within range) are more similar between closely related species than expected by chance, we tested for the presence of phylogenetic signal in these attributes. Testing for such phylogenetic signal can help us interpreting observed patterns, in this case the relationship between two species' attributes (e.g., range size ~ latitudinal position), under a phylogenetic perspective about each individual attribute [82]. For this, we calculated phylogenetic signal for each species' attribute using conventional metrics such as Blomberg's K and Pagel's lambda with the "phylosig" function from the "phytools" package [83].

**2.5.2 Ordinary least squares.** We explored the validity of Rapoport's rule and evaluated the environmental hypotheses using ordinary least squares (OLS) models. Unlike Rapoport's rule validation, where we used univariate regressions (as latitude was our single predictor

variable), for the evaluation of environmental hypotheses we conducted multiple regressions. This approach allowed us to consider the effect of all explanatory variables within the model while measuring how much variability in the data can be explained by each variable [84]. To do this, we employed a multi-model inference approach based on information theory [85]. We constructed multiple models that considered all possible combinations of our predictor variables using the "dredge" function from the "MuMIn" package [86]. The relative support for each model was determined based on their Akaike weights. At the cross-species level, we found limited support for most variable combinations, where all combinations of variables had a similar probability of being chosen as the best model (S1 Appendix). Therefore, we performed coefficient averaging using the 95% of the Akaike weights as confidence set [85]. This was performed using the "model.avg" function from the "MuMIn" package [86]. At the assemblage level, the best model consistently included all predictors and had strong support (high Akaike weights, S1 Appendix). Thus, model averaging was not performed.

**2.5.3 Phylogenetic generalized least squares.** When analyzing relationships at the cross-species level, it is possible that species share similar attribute values simply because they are phylogenetically related. Therefore, we need to account for the phylogenetic non-independence among species, as it violates the assumption of independence among observations [87]. To do this, we employed the Phylogenetic generalized least squares (PGLS) method to incorporate the phylogenetic autocorrelation within the error structure of our regressions and thus avoid type 1 error [88]. To do so, we used the "pgls" function from the "caper" package [89]. Then, we performed the same model selection and averaging procedure described for OLS.

**2.5.4 Simultaneous autoregressive models.** Due to the potential presence of spatial autocorrelation in the assemblage level approach (where close sites may share similar climates and similar range size values), we performed simultaneous autoregressive models (SARs) to directly consider this autocorrelation, using the "errorsarlm" function from the "spatialreg" package [90]. These models are used to incorporate spatial autocorrelation in statistical analyses, using neighborhood matrices to define the relationship between the model residuals at each location and those of neighboring locations [91]. For the environmental hypotheses, we selected the model with the best support based on OLS (which considered all predictor variables). In order to determine the most appropriate spatial weights matrix, we followed a similar protocol to the one used in Skeels *et al.* [92]. For this, we constructed four different models (one OLS and three SARs). The neighborhood distance matrix used was 50 km (distance between the centroids of the cells). Three schemes were applied for this distance matrix [93]: One row standardized, another globally standardized, and a variance stabilizing. The selection of the best model was based on comparing the lowest values of the Akaike Information Criterion and the highest values of $R^2$ (S2 Appendix).

## 2.6 Null models

To compare our results against expected relationships if climate did not affect range size variation, we created a series of null models. These null models assumed that the spatial distribution of range size is not influenced by environmental gradients and depending only on area availability [9]. Our null hypothesis was based on cohesive range models using the spreading-dye algorithm [94]. Using the previously generated presence-absence matrix, we applied the "rangemod2d" function from the "RangemodelR" package to randomize the position of species ranges while preserving their size [28]. This function is based on the simulation implemented in Rahbek *et al.* [95], where for each species, an initial site of the domain is randomly selected, and then neighboring cells are added until the number of occupied grid-cells for the focal species is reached. This procedure was repeated 100 times. Then, we calculated the expected

regression coefficients (slopes) for OLS, PGLS, and SARs using the procedure described for the observed data. Null frequency distributions were generated from the 100 simulated coefficients to compare with the observed coefficients. We considered the observed coefficients to be significantly different from the null coefficients if they fell outside the 95% confidence interval of their frequency distributions. Given the directional nature of our hypotheses (i.e., expecting positive relationships for some hypotheses and negative relationships for others), we decided to use a one-tailed test. Thus, for those associated with negative predictions, we opted for the 0.05 percentile from the null distribution to establish the cutoff threshold. Conversely, for hypotheses linked to positive predictions, we utilized the 0.95 percentile.

## 3. Results

### 3.1 Cross-species level analysis

We detected a higher number of species with small range sizes creating a left-skewed distribution and a log-normal distribution was observed when the range size was log-transformed (Fig 1). The range sizes varied by up to four orders of magnitude. The smallest range sizes were 6,160.5 km$^2$ for *S. esperanzae* and *S. macdougalli*, and the largest range size was 5,436,641.25 km$^2$ for *S. undulatus* (S2 Table).

Using Pagel's lambda, significant phylogenetic signals were observed for all attributes except for range size ($\lambda = 0$ and p = 1) regardless of the phylogeny used. Similarly, Blomberg's K analysis revealed phylogenetic signals for all attributes except for range size and climate change velocity in both phylogenies (Table 2).

Our results at the cross-species level indicated support for Rapoport's rule in all three datasets (Fig 2 and Table 3), confirming a positive relationship between range size and latitude (Slope = 0.5–0.53, R$^2$ = 0.24–0.27, p < 0.01). The results for OLS and PGLS were very similar,

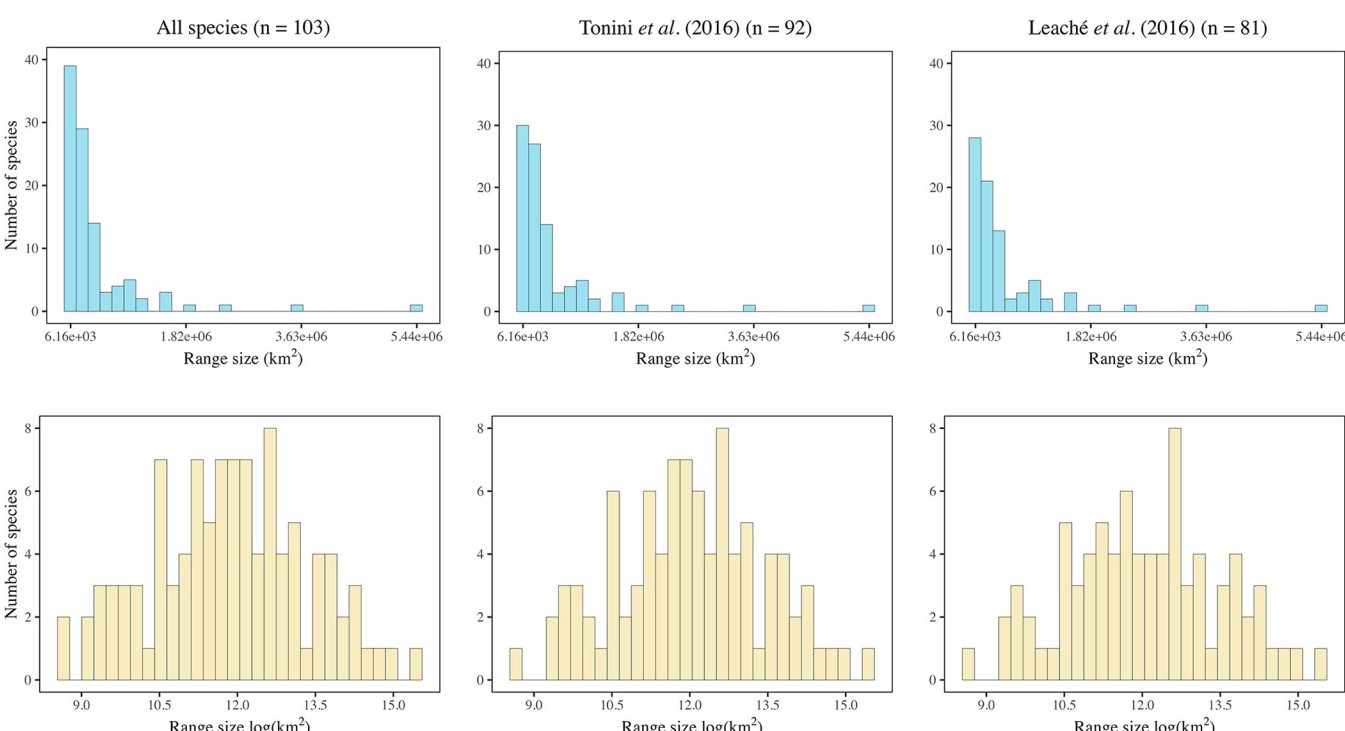

**Fig 1.** Frequency distribution of range sizes in km$^2$ (left) and natural logarithm of range size (right) at cross-species level across all datasets.

**Table 2. Phylogenetic signal for all the attributes across Leaché *et al*. [38] and Tonini *et al*. [39] datasets.** Bold values indicate significant results.

| Trait | K | p-value | λ | p-value | Phylogeny |
|---|---|---|---|---|---|
| Latitudinal midpoints | **0.46** | **< 0.01** | **0.79** | **< 0.01** | Tonini *et al*. [39] |
| log (Range size) | 0.24 | 0.09 | 0 | 1 | |
| Climate change velocity | 0.23 | 0.32 | **0.35** | **< 0.05** | |
| Elevation | **0.26** | **< 0.05** | **0.38** | **< 0.05** | |
| Minimum temperature of the coldest month | **0.35** | **< 0.01** | **0.66** | **< 0.01** | |
| Temperature annual range | **0.38** | **< 0.01** | **0.70** | **< 0.01** | |
| Latitudinal midpoints | **0.77** | **< 0.01** | **0.98** | **< 0.01** | Leaché *et al*. [38] |
| log (Range size) | 0.28 | 0.37 | 0 | 1 | |
| Climate change velocity | 0.37 | 0.10 | **0.56** | **< 0.01** | |
| Elevation | **0.49** | **< 0.01** | **0.70** | **< 0.01** | |
| Minimum temperature of the coldest month | **0.63** | **< 0.01** | **0.91** | **< 0.01** | |
| Temperature annual range | **0.71** | **< 0.01** | **0.98** | **< 0.01** | |

and no phylogenetic signal was observed for the range size-latitude relationship in any dataset (λ = 0, Table 3). Regarding the environmental hypotheses, the model including only minimum temperature of the coldest month and elevation was selected as the best model, although it had low support (Akaike weights = 0.17–0.20; S1 Appendix). Still, the higher importance of these two variables, compared to the other ones, was confirmed in the model averaging approach, where the climatic extremes hypothesis consistently had the stronger support (Slope = -0.45 to -0.33) followed by the elevation hypothesis (Slope = -0.14 to -0.18). Given this model averaging, the other hypotheses were also considered with positive relationships with climate change velocity (Slope = 0.13–0.16) and temperature annual range (Slope = 0.04–0.17), although these had less relevance in explaining range size variation across space (Table 3).

### 3.2 Assemblage level analysis

Our results at this level also provided support for Rapoport's rule, where in average, we observed smaller range sizes in tropical regions and larger range sizes in temperate regions for the complete dataset (Fig 3). This was true regardless of the analyzed dataset (Fig 4 and Table 4), with positive slopes in the relationship between range size and latitude (Slope = 0.68–0.72, $R^2$ = 0.5–0.96, p = 0). In general, SARs had higher $R^2$ values (0.96 in all three datasets) compared to OLS (0.5–0.51). The slopes were similar between OLS and SARs, with SARs

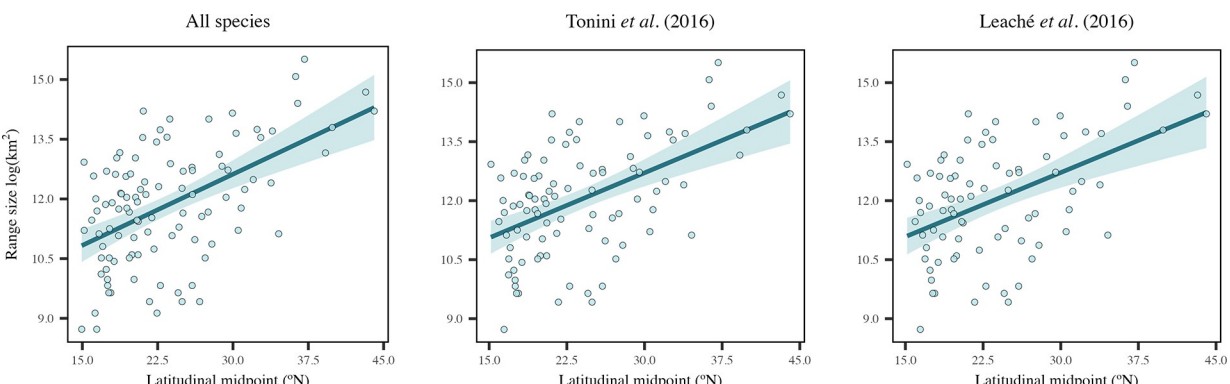

**Fig 2. Relationship between range size and latitudinal midpoints for the three datasets (all species, Leaché *et al*. [38], and Tonini *et al*. [39]) at the cross-species level.**

**Table 3. Regression parameters for Rapoport's rule and model-averaged slopes of environmental hypotheses at the cross-species level.** OLS = Ordinary Least Squares, PGLS = Phylogenetic Generalized Least Squares. CEH = Climatic Extremes Hypothesis, EH = Elevation Hypothesis, HCSH = Historical Climatic Stability Hypothesis, CVH = Climatic Variability Hypothesis.

| | All species | Tonini *et al.* [39] | | Leaché *et al.* [38] | |
|---|---|---|---|---|---|
| | **OLS** | **OLS** | **PGLS** | **OLS** | **PGLS** |
| | Rapoport's rule | | | | |
| Slope | 0.53 | 0.52 | 0.52 | 0.50 | 0.50 |
| $R^2$ | 0.27 | 0.26 | 0.26 | 0.24 | 0.24 |
| $\lambda$ | - | - | 0 | - | 0 |
| p-value | < 0.01 | < 0.01 | < 0.01 | < 0.01 | < 0.01 |
| | Environmental hypotheses | | | | |
| CEH | -0.45 | -0.33 | -0.33 | -0.36 | -0.36 |
| EH | -0.18 | -0.16 | -0.16 | -0.14 | -0.14 |
| HCSH | 0.15 | 0.13 | 0.13 | 0.16 | 0.16 |
| CVH | 0.04 | 0.17 | 0.17 | 0.08 | 0.08 |

having slightly lower values (Slope = 0.68–0.70, Table 4). The climatic extremes hypothesis was the most relevant in explaining the variation in range size across all datasets ($Slope_{OLS}$ = -0.85 to -0.80 and $Slope_{SARs}$ = -0.35 to -0.30) followed by the elevation hypothesis ($Slope_{OLS}$ = -0.42 to -0.43 and $Slope_{SARs}$ = -0.14 to -0.12) and historical climatic stability hypothesis ($Slope_{OLS}$ = 0.21–0.22, $Slope_{SARs}$ = 0.02). The climatic variability hypothesis was not supported at the assemblage level, as we observed low negative slopes (Table 4).

## 3.3 Null models

Regarding the null models, for Rapoport's rule, we found significant differences between the observed coefficient and the simulated ones at both the cross-species and assemblage levels in all datasets, regardless of the statistical method used to analyze the relationship (S2 Fig and Fig 5). Concerning the environmental hypotheses, for the OLS models (S2 Fig), the observed coefficients for climatic extremes and elevation hypotheses were significantly different from the null coefficients in all datasets, both at the cross-species and assemblage levels. On the other hand, the observed slopes for the historical climatic stability and climatic variability hypotheses did not differ from the null values in any dataset or level of analysis (S2 Fig). For PGLS, only the climatic extremes and elevation hypotheses significantly differed from the null values (Fig 5). Regarding the SARs, we did not find significant differences between the observed and simulated coefficients for any of the environmental hypotheses (Fig 5). The individual coefficients of each simulated regression model can be found in S3 and S4 Appendices.

## 4. Discussion

In this study, we evaluated the application of Rapoport's rule in spiny lizards (genus *Sceloporus*) and assessed the importance of multiple environmental hypotheses to explain their latitudinal range size variation. Our results confirmed a positive relationship between range size and latitude, providing support for Rapoport's rule in this lizard genus. This finding is consistent with previous studies supporting the validity for this rule in the analyzed region [6, 8, 11, 96], as well as in other reptile groups such as snakes [20, 97]. However, studies evaluating Rapoport's rule in lizards are still limited. Cruz *et al.* [16] found a positive relationship between range size and latitude when analyzing a small sample of lizards belonging to the *Liolaemus* genus. Conversely, Pincheira-Donoso [17], observed an inverse pattern to that predicted by

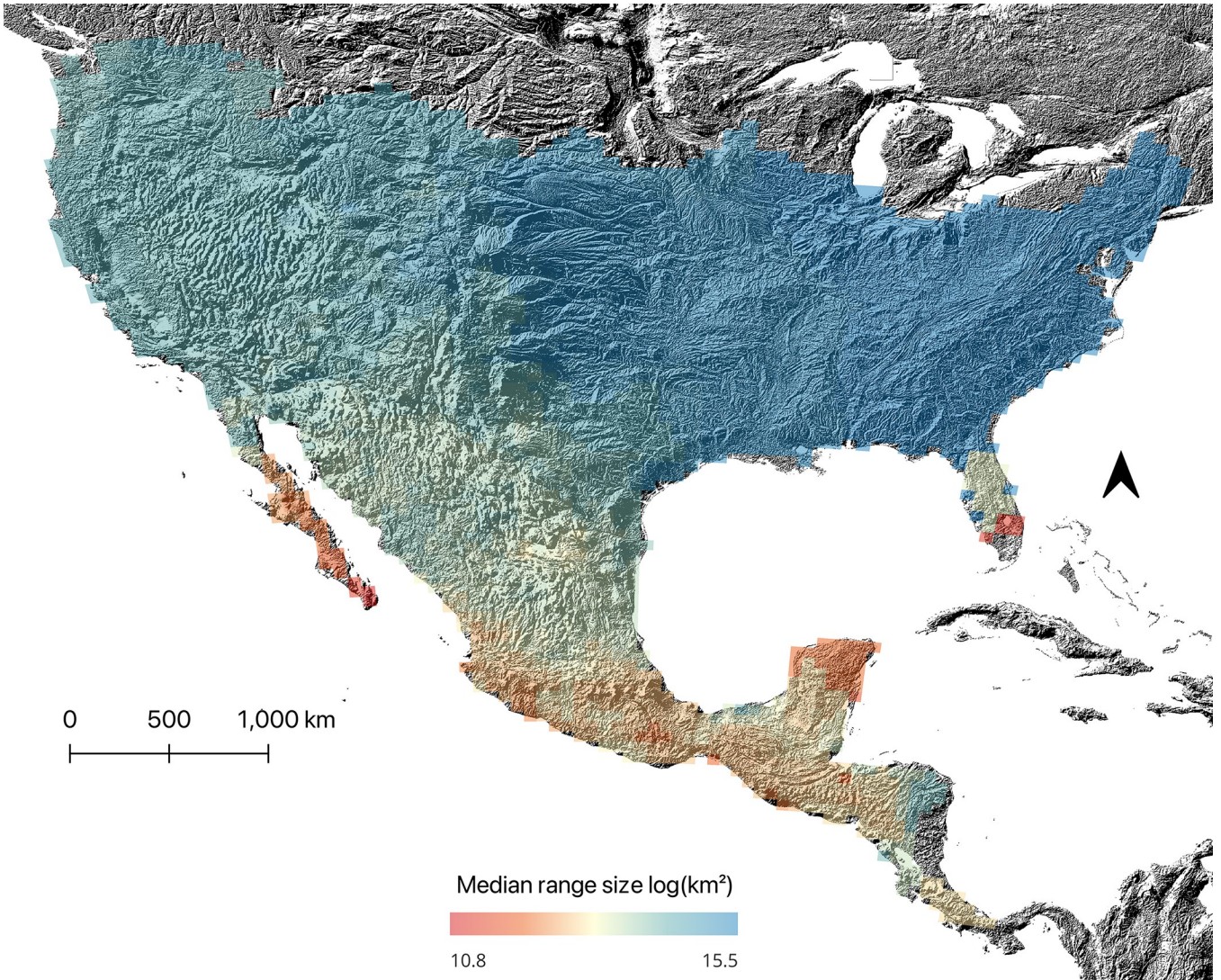

**Fig 3. Geographic pattern of median (log) range size of *Sceloporus* species.** Map projection is in Albers equal area. This map was created in QGIS 3.32 [50]. The elevation base map was derived from SRTM data of the USGS Earth Resources Observatory and Science (EROS) center (public domain: https://eros.usgs.gov#) as provided by WorldClim 2.1 [47] under CC-BY license.

Rapoport's rule for the same genus when analyzing a larger number of species. Moreover, Pintor *et al*. [98] found mixed results when analyzing Rapoport's rule in three different clades of lizards in Australia (*Carlia*, *Ctenotus* y *Egernia*). These findings emphasize that different patterns can emerge when analyzing different clades or number of species [16]. Despite this, our results were consistent regardless of our analyzed datasets thus supporting the validity of Rapoport's rule in the *Sceloporus* genus.

Our confirmation for the application of Rapoport's rule was also consistent both at the cross-species and assemblage levels, as has been found in other taxa [8, 20, 97, 98]. This happens because both approaches are grounded in the same underlying principle, where the assemblage of species emerges from the overlap of their individual distributions [99]. Therefore, when the majority of species with large range sizes are found at temperate latitudes, the overlapping of their distributions lead to those locations having, on average, larger range sizes compared to tropical regions.

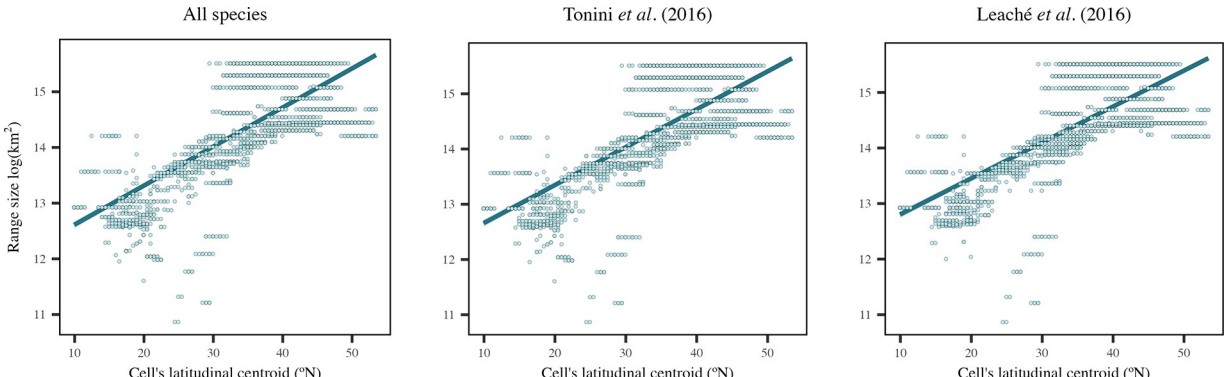

**Fig 4. Relationship between range size and cell's latitudinal centroid for the three datasets (all species, Leaché *et al.* [38], and Tonini *et al.* [39]) at the assemblage level.**

Concerning the evaluated environmental hypotheses, our results showed consistency between the cross-species and the assemblage level analyses. Overall, the climatic extremes hypothesis consistently had the highest support, in line with findings from previous studies conducted in both plants and animals [21, 100, 101]. This hypothesis suggests that species capable of tolerating more extreme temperature limits, are able to expand the boundaries of their geographic ranges [21]. In accordance with this, studies conducted in *Sceloporus*, found that species that tolerate colder temperatures tend to have larger geographic ranges [102]. Likewise, studies conducted on other genus of lizards such as *Liolaemus* showed that cold climate species exhibit greater physiological plasticity and have larger geographic ranges [103]. Within *Sceloporus*, species capable of tolerating low temperatures, such as those found in temperate regions, exhibit greater efficiency in thermoregulation, maintaining their body temperature even in cold climates by employing ethological strategies like limiting their activity season through hibernation [37]. Both characteristics make it easier for them to spread in colder climates.

These observations are closely related to elevation which was another important factor in determining the spatial variation of *Sceloporus* range sizes. According to the seminal hypothesis proposed by Janzen [25], mountains in tropical regions represent a more restrictive physiological barrier than in temperate regions. If this mechanism is operating to limit range size,

**Table 4. Regression parameters for Rapoport's rule and best supported model slopes of environmental hypotheses at the assemblage level.** OLS = Ordinary Least Squares, SARs = Simultaneous Autoregressive models. CEH = Climatic Extremes hypothesis, EH = Elevation Hypothesis, HCSH = Historical Climatic Stability Hypothesis, CVH = Climatic Variability Hypothesis.

| | All species | | Tonini *et al.* [39] | | Leaché *et al.* [38] | |
|---|---|---|---|---|---|---|
| | **OLS** | **SARs** | **OLS** | **SARs** | **OLS** | **SARs** |
| | Rapoport's rule | | | | | |
| Slope | 0.72 | 0.68 | 0.71 | 0.68 | 0.71 | 0.70 |
| R² | 0.51 | 0.96 | 0.50 | 0.96 | 0.50 | 0.96 |
| p-value | 0.00 | 0.00 | 0.00 | 0.00 | 0.00 | 0.00 |
| | Environmental hypotheses | | | | | |
| CEH | -0.84 | -0.31 | -0.85 | -0.30 | -0.80 | -0.35 |
| EH | -0.43 | -0.14 | -0.42 | -0.12 | -0.42 | -0.13 |
| HCSH | 0.21 | 0.02 | 0.22 | 0.02 | 0.22 | 0.02 |
| CVH | -0.08 | -0.08 | -0.09 | -0.07 | -0.05 | -0.05 |

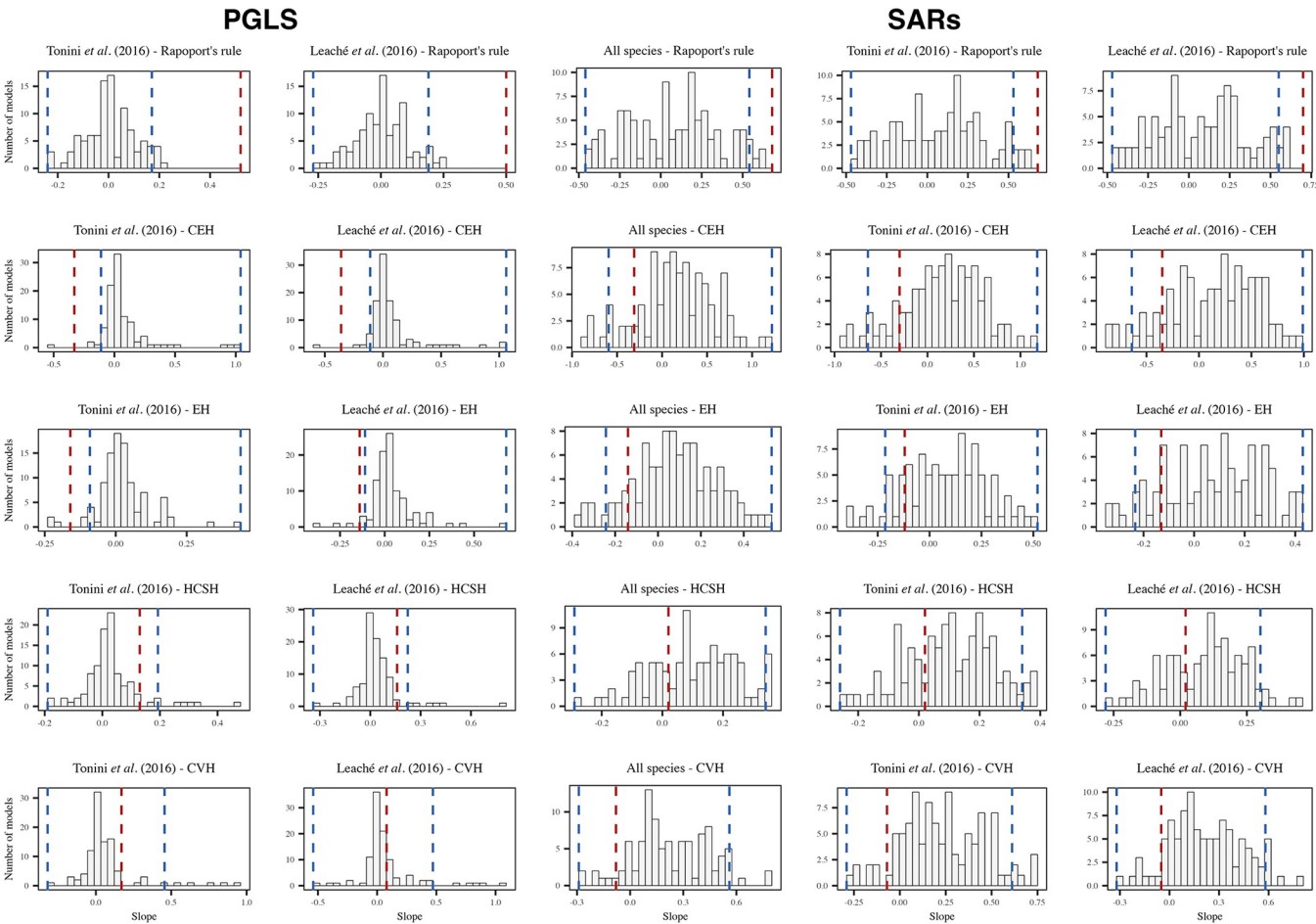

**Fig 5. Frequency distributions of the null slopes across 100 regression models using simultaneous autoregressive models and phylogenetic generalized least squares.** Red dashed line = Observed value. Blue dashed lines = 0.05% and 95% interval confidence of the null distribution. CEH = Climatic Extremes Hypothesis, EH = Elevation Hypothesis, HCSH = Historical Climatic Stability Hypothesis, CVH = Climatic Variability Hypothesis.

species in these regions should exhibit high genetic differentiation among their populations, speciation through allopatry, and high species richness in tropical mountain regions [104]. Indeed, for the genus *Sceloporus*, there is evidence suggesting that these three processes may be occurring across mountain regions. Grummer *et al.* [105] observed limited genetic flow and genetic differentiation among populations of mountain-dwelling species within the *Sceloporus spinosus* group. Furthermore, allopatric speciation has been previously suggested as one of the most plausible explanations for the diversification of this lizard group [45, 106]. Additionally, it has been observed for this genus that tropical regions with high topographic variation are associated with high species richness [45]. Likewise, it has been observed that *Sceloporus* species in temperate regions are able to maintain their body temperature regardless of elevation, contrasting with tropical zones where mountain species differ in body temperature compared to lowland species [37]. Previously, in other groups of lizards (families Scincidae and Varanidae), it was also observed that elevation had a negative effect on species' dispersal ability [107], and high-elevation areas are associated with low genetic diversity and small population sizes [108]. Additional studies have also found that elevation affects latitudinal patterns of range size in other ectotherm taxa [15, 18].

Similarly, we also found support for the remaining hypotheses, but with slight relevance in explaining the observed spatial pattern in range size. Regarding the climatic variability hypothesis, our results contrast with the original proposal of Stevens [5], who suggested this hypothesis as one of the main explanations for Rapoport's rule. Although it has received broad support in explaining spatial variation in range size in different groups of reptiles such as snakes [20] and some lizard clades [16, 89], other studies have rejected this hypothesis as the primary determinant of Rapoport's rule even for the same lizard clades (*Liolaemus*) [103, 109]. In particular for *Sceloporus* lizards, the lack of support for the climatic variability hypothesis may be due to their ability to maintain their body temperatures across seasons throughout the year, thus not being affected by such temperature variability in determining their movements and thus their range sizes [110].

We also observed that the historical climatic stability hypothesis received statistical support but had little power to explain Rapoport's rule for our lizard genus. One of the main drivers behind this hypothesis is the extinction of small-ranged species at northern latitudes due to climatic instability [24]. Thus, our results suggests that extinction processes may not have been a significant factor in range size dynamics of *Sceloporus* lizards. Indeed, for this genus, low and stable extinction rates have been consistently observed throughout its evolutionary history [38]. Moreover, sites with the highest species diversity for the genus [45] coincide with regions where past climate variation has been relatively stable, such as the center of Mexico [24].

Although some environmental variables were more important than others in explaining the range size spatial pattern of *Sceloporus*, all variables contributed to our final models. First, this implies that no single hypothesis can solely explain the causes behind Rapoport's rule in this lizard clade. Still, among the evaluated hypotheses, some had more support (climatic extremes and elevation) than others, which allowed us to infer the most important drivers of this ecogeographical rule in our studied group. This was also consistent between cross-species and assemblage levels, where averaged model and best model, respectively, showed higher importance of minimum temperature of the coldest month and elevation. Models for both levels exhibited a good fit to the data, explaining a significant amount of variance of the range size spatial pattern. Considering our model selection approach, this is relevant because it provides support to our inferences beyond simply relying on a best model that could perform better than a set of candidates but still provide poor explanatory power [111, 112].

Our analyses also revealed a correspondence between phylogenetically related species and their geographic and environmental proximity. Accordingly, our results showed phylogenetic signal in the location of species (i.e., their latitudinal midpoint) as well as in their occupied environments as shown by most of our evaluated variables. However, range size did not show phylogenetic signal, which supports the idea that speciation within *Sceloporus* primarily took place through allopatric processes [45, 106], since this event can disrupt the preservation of range size while still keeping species in close environmental and geographic proximity [113]. This finding provides a clear explanation for the lack of phylogenetic signal in the relationship between range size and both latitude and climate.

Given that our basic study unit, species' geographic ranges, was based on open data available from secondary sources such as GBIF and Naturalista, it is important to note the limitations and potential biases of this data. Some limitations of data from this sources includes: 1) occurrence records may be spatially autocorrelated, which can affect the construction of potential distributions using correlative approaches by, for example, estimating greater climate suitability towards heavily sampled regions [114]; 2) occurrence records often have identification and georeferencing issues [115], potentially resulting in records outside the known species' range, thus leading to range size overestimations; and 3) some records may come from suboptimal environmental conditions (e.g., sink populations), potentially affecting the

estimated climatic conditions occupied by species and thus biasing the evaluation of environmental hypotheses. We tried to avoid these biases as much as possible by conducting a thorough cleaning process, but some inherent biases could still remain present. Nevertheless, previous studies have demonstrated the feasibility of accurately estimating both species range sizes and climatic variability/extremes using this type of secondary biodiversity data (e.g., [116, 117]), thus supporting the use of these repositories in our study.

When we contrasted our results with null scenarios, we discovered that the spatial variation of range size for the genus *Sceloporus* differs significantly from null expectations regardless of the level of analysis. This finding indicates that our observed patterns cannot be replicated by randomizing the location of species' ranges under the assumption of no environmental control. Therefore, other factors more than geometric restrictions should be influencing their spatial variation in range size. However, for the environmental hypotheses, we observed differences between the levels of analysis. At the cross-species level, for the climatic extremes hypothesis we observed significant deviations from what was expected based on the null model. This can be explained by the fact that the study region exhibits a pronounced latitudinal temperature gradient, with colder areas in the temperate region [47]. Therefore, randomizing the geographic position will not necessarily replicate the relationship found in the observed data. For instance, it is possible that small range size values are randomly placed in extremely cold locations, which would generate different relationships than the observed pattern. Likewise, we confirm that the relationship between range size and elevation follows a non-random pattern. The presence of this signal can be explained by the fact that, in tropical areas, species face greater difficulty in migrating between regions with different elevations [25, 104]. On the contrary, despite the presence of diverse mountain ranges in the northern latitudes of our study region [118], species distributions in these areas also extend across the vast plains of North America. This occurrence results in their distribution having, on average, sites of intermediate elevation, distinguishing their tropical counterparts. Nevertheless, null models have the capability to position species (regardless of their range size) in regions of high or low elevation indiscriminately, resulting in relationships that diverge from those observed in the real data.

Conversely, the observed relationships for the climatic variability hypothesis and the historical climatic stability hypothesis were not significantly different from the null expectations. This result makes sense due to the low contribution of these variables in explaining range size, so it is not surprising that their coefficients can be replicated by our simulations. At the assemblage level, our analyses did not reveal significant differences compared to the null models for any of the studied hypothesis. Previous studies have emphasized that spatial autocorrelation can produce significant relationships between climate and range size [29, 119]. Indeed, these relationships can originate from the inherent spatial structure of climate variables and species' attributes at the assemblage level, such as range size [120, 121]. This latter phenomenon arises because the range size of an assemblage emerges from the overlapping of individual species, leading to issues of pseudo-replication [120]. Additionally, when combined with the pronounced spatial structure present in climatic variables, these factors can contribute to the emergence of strong associations that lack biological meaning. Moreover, it is also possible that the overlap of species' ranges may be influenced by other factors beyond climate, such as biotic interactions [119]. Thus, while we observed a strong association with certain environmental variables and range size at the assemblage level, we were unable to distinguish such association from the effect of geometric constraints and inherent spatial structure. Still, pseudo-replication issues and spatial structure are less important at the cross-species level, which supports our overall findings and emphasizes the need to consider both levels of analyses for a robust evaluation of ecogeographical patterns [81].

## 5. Conclusions

We found support for Rapoport's rule at both the cross-species and assemblage levels in *Sceloporus*. Even though our region of analysis has strong geometric restrictions in landmasses, which would make it prone to exhibiting Rapoport's rule, our null models, which are useful for detecting these effects, revealed significant differences between the observed and simulated relationships. This indicates that the Rapoport's rule confirmed in our study is not a consequence of such constraints. In general, the climatic extremes and elevation hypotheses had the highest support. At the cross-species level, these observations were significantly different from what was expected by null simulations, suggesting that this effect is linked to the physiology of the species. At the assemblage level, we did not observe significant differences from the null models for any environmental hypotheses, showing a strong effect of spatial structure in shaping these relationships or the possible influence of other factors affecting the species assemblage.

## Supporting information

**S1 Appendix. Results of model selection based on Akaike Information Criterion for environmental hypotheses.** This table contains the raw values from all the combinations between environmental hypotheses used for model selection.
(PDF)

**S2 Appendix. Results of simultaneous autoregressive models.** This appendix shows the results of the simultaneous autoregressive models for Rapoport's rule and environmental hypotheses.
(PDF)

**S3 Appendix. Coefficients for null simulations for Rapoport's rule.** These tables contain all the simulated coefficients for the relation between range size and latitude.
(PDF)

**S4 Appendix. Coefficients for null simulations for environmental hypotheses.** These tables contain all the simulated coefficients for the relation between range size and every environmental hypothesis.
(PDF)

**S1 Fig. Range size and midpoints' comparisons for all methods.** This figure includes the comparation for range size and midpoints estimations for the three methods employed in our study (Alpha-hull, Convex-hull and Species Distribution Models).
(PDF)

**S2 Fig. Frequency histograms for null OLS (cross-species and assemblage level).** This figure shows the 100 simulated OLS coefficients at cross-species and assemblage level.
(PDF)

**S1 Table. Number of occurrence records per species.** This table contains the number of occurrence records used to model the distribution ranges.
(PDF)

**S2 Table. Range size estimated with alpha hulls for all the species.** This table contains the range size estimation for all the species through alpha-hulls.
(PDF)

## Acknowledgments

To the Posgrado en Ciencias Biológicas (PCB) at the Universidad Nacional Autónoma de México (UNAM), since this work is a derivative of the doctoral research of KLR in this program, and this article fulfills one of the requirements for academic degree. We thank Dr. Luis Daniel Ávila Cabadilla, Dr. Enrique Martinez-Meyer, and Dr. Julián Velasco for comments and discussions that help improve this work. We also thank the anonymous reviewers for their valuable comments that helped improve the quality of this work. Finally, to the members of the evolutionary macroecology lab (https://maevolab.mx) for the support in phylogenetic and statistical analysis during this research.

## Author Contributions

**Conceptualization:** Kevin López-Reyes, Carlos Yáñez-Arenas, Fabricio Villalobos.

**Data curation:** Kevin López-Reyes.

**Formal analysis:** Kevin López-Reyes, Fabricio Villalobos.

**Funding acquisition:** Kevin López-Reyes, Carlos Yáñez-Arenas.

**Investigation:** Kevin López-Reyes, Carlos Yáñez-Arenas, Fabricio Villalobos.

**Methodology:** Kevin López-Reyes, Fabricio Villalobos.

**Project administration:** Carlos Yáñez-Arenas, Fabricio Villalobos.

**Resources:** Kevin López-Reyes, Carlos Yáñez-Arenas.

**Software:** Kevin López-Reyes, Fabricio Villalobos.

**Supervision:** Carlos Yáñez-Arenas, Fabricio Villalobos.

**Validation:** Kevin López-Reyes, Fabricio Villalobos.

**Visualization:** Kevin López-Reyes.

**Writing – original draft:** Kevin López-Reyes.

**Writing – review & editing:** Kevin López-Reyes, Carlos Yáñez-Arenas, Fabricio Villalobos.

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
