## [Decision Letter · Decision Letter 0]

29 Jan 2024

PONE-D-23-29199Exploring the causes underlying the latitudinal variation in range sizes: evidence for Rapoport’s rule in spiny lizards (genus Sceloporus)PLOS ONE

Dear Dr. López Reyes,

Thank you for submitting your manuscript to PLOS ONE. After careful consideration, we feel that it has merit but does not fully meet PLOS ONE’s publication criteria as it currently stands. Therefore, we invite you to submit a revised version of the manuscript that addresses the points raised during the review process. Since the MS is reviewed by three expert reviewers, please pay special attention to the comments of all the reviewers during the revision process. Please submit your revised manuscript by Mar 14 2024 11:59PM. If you will need more time than this to complete your revisions, please reply to this message or contact the journal office at plosone@plos.org. Please include the following items when submitting your revised manuscript:A rebuttal letter that responds to each point raised by the academic editor and reviewer(s). You should upload this letter as a separate file labeled 'Response to Reviewers'.A marked-up copy of your manuscript that highlights changes made to the original version. You should upload this as a separate file labeled 'Revised Manuscript with Track Changes'.An unmarked version of your revised paper without tracked changes. You should upload this as a separate file labeled 'Manuscript'.

We look forward to receiving your revised manuscript.

Kind regards,

Bhoj Kumar Acharya, PhD

Academic Editor

PLOS ONE

Reviewers' comments:

Reviewer's Responses to Questions

**Comments to the Author**

1. Is the manuscript technically sound, and do the data support the conclusions?

Reviewer #1: Partly

Reviewer #2: Yes

Reviewer #3: Partly

2. Has the statistical analysis been performed appropriately and rigorously? 

Reviewer #1: Yes

Reviewer #2: Yes

Reviewer #3: Yes

3. Have the authors made all data underlying the findings in their manuscript fully available?

Reviewer #1: No

Reviewer #2: Yes

Reviewer #3: Yes

4. Is the manuscript presented in an intelligible fashion and written in standard English?

Reviewer #1: Yes

Reviewer #2: Yes

Reviewer #3: Yes

5. Review Comments to the Author

Reviewer #1: Dear Authors,

I found it as interesting work exploring the latitudinal range size patterns of a least studied taxa. I have a few comments/concerns before going to the next step.Please find the attached document for my comments. Thank you and looking forward to see this get published.

Reviewer #2: It's great to see authors dive into an in-depth analysis to further validate Rapport's rule, especially focusing on latitudinal variation in range sizes using spiny lizards as a model organism. There's indeed a lack of analytical information regarding latitudinal range size variation in lizards, making this study particularly interesting.

The manuscript appears well-written and offers a robust evaluation of four ecological hypotheses, namely historical climate stability, climate variability, climate extremes, and elevation. It's worth noting that the findings strongly support the climate extremes and elevation hypotheses as the primary drivers of spatial variation in range size, surpassing the other two hypotheses.

Nonetheless, it's essential to acknowledge that the reliance on secondary data from sources like the Global Biodiversity Information Facility and Naturalista comes with inherent limitations and potential biases. This is a crucial aspect to consider in the interpretation of the results.

Regarding the methodology section, while it's good that appropriate statistical models were used, it's important to strike a balance between the statistical details and the ecological context. Enhancing the clarity of the methodology section could make the paper more engaging and accessible to a broader audience, reducing the perception of it being solely a statistical manuscript.

Overall, it's a commendable effort in shedding light on this underexplored area of research. Kudos to the authors for their dedication to advancing our understanding of range size variation in lizards.

Reviewer #3: Phylogenetic part needs a major revision

The information provided for the phylogenetic reconstruction and divergence dating is severely inadequate. There are multiple details that need to provided. Please refer to the comments provided in the manuscript.

6. PLOS authors have the option to publish the peer review history of their article (what does this mean?). If published, this will include your full peer review and any attached files.

Reviewer #1: No

Reviewer #2: **Yes: **Ramesh Chinnasamy

Reviewer #3: No

---

## [Author Response · Author response to Decision Letter 0]

27 Apr 2024

PONE-D-23-29199 - Response Letter

Editor Comments:

Thank you for submitting your manuscript to PLOS ONE. After careful consideration, we feel that it has merit but does not fully meet PLOS ONE’s publication criteria as it currently stands. Therefore, we invite you to submit a revised version of the manuscript that addresses the points raised during the review process. Since the MS is reviewed by three expert reviewers, please pay special attention to the comments of all the reviewers during the revision process.

R: We appreciate the consideration of our manuscript and the opportunity to revise it following the careful editorial and review process. Below, we provide a point-by-point response (in plain font) to all of the reviewers’ comments and suggestions (in italics). 

Reviewer comments:

Reviewer #1

Dear Authors, I found it as interesting work exploring the latitudinal range size patterns of a least studied taxa. I have a few comments/concerns before going to the next step. Please find the attached document for my comments. Thank you and looking forward to see this get published.

R: We thank the reviewer for the positive comments and useful suggestions that have helped us to improve our manuscript. 

1.1. It is really an interesting work exploring species range sizes along latitude. Also, ectothermic and least studied groups like reptiles (especially lizards) are the best models for such experiments. With my limited knowledge on these analytical approaches, I feel like the analysis is more like a test of Climatic Variability Hypothesis rather than Rapoport’s rule. Because, the analyses conducted in this paper first explore the geographical range of each species using SDM or extent of occurrences (polygon methods), then modeling these range sizes with variability in climatic variables (current and historic) and elevation. In method, use of ‘range size in km2 (response variable) and latitudinal midpoint (explanatory variable) shows that the overall geographical range size is taken as the response variable instead of the latitudinal range sizes (or range limits).

The (latitudinal) Rapoport’s rule is a test of latitudinal range sizes of species (not the overall

geographical distribution range size) with their mid-point distribution across latitude. Meanwhile Climatic Variability Hypothesis explore multiple dimensions of species ranges with respect to climatic variability in any geographical space (latitudinal/ altitudinal) as it is done in this paper. If this is correct, I would request authors to set the objectives of the paper in that aspect.

R: We understand the reviewer’s concern and appreciate the opportunity to clarify our approach. Indeed, the original proposal of Rapoport’s rule made by Stevens (1989) considered latitudinal extent as the response variable for two main reasons: 1) because it yielded a clearer correlation with latitude than using areal extent (i.e., actual size or surface in squared units) and 2) because it better reflects the climatic challenges faced by species. However, ever since the classic studies evaluating Rapoport’s rule under this ‘one-dimensional’ perspective using latitudinal extent (reviewed in Ruggiero & Werenkraut, 2007), it has become evident that patterns vary in multiple dimensions and that ‘latitude’ alone does not provide a truly geographical approach (Hawkins & Diniz-Filho, 2004; Ruggiero & Hawkins, 2006). As such, geographical data should ideally be analyzed in the same dimensional space in which they occur (Hawkins & Diniz-Filho, 2006). Accordingly, most of the recent studies evaluating Rapoport’s rule now consider areal extent or size instead of simply latitudinal extent to evaluate the validity of the rule and its underlying causes (Lou et al. 2011; Ten Caten et al. 2023). We now include this explanation in our revised methods section (please, see lines 164-170).

In fact, considering the actual areal/size of species’ geographic ranges allows to describe the full climatic conditions that they occupied and evaluate their influence in determining the proposed rule of a latitudinal gradient in geographic range size as originally suggested by Rapoport (1975) and formalized by Stevens (1989) (e.g., Pintor et al. 2015; cited by the reviewer and originally included in our references). This is what we aimed with our study, to both describe the existence of Rapoport’s rule in Sceloporus lizards considering different methods to construct geographic ranges and evaluate the main hypotheses posited to explain such pattern. To avoid potential confusion and provide a clearer message to the reader, we have now rephrased our goals to explicitly state our two-fold objective of testing for Rapoport’s rule and evaluating its proposed environmental drivers (for example, see lines 67-90 and 122-133).

1.2. Apart from the above, I would also like to see the sample sizes which is the number of occurrence points used for modeling each species in the analysis. In methods the authors mentioned that species with at least three validated points were also used. I am not sure if such a small number of points can define the climatic variables or range sizes (or if it is sufficient for SDMs) of a species.

R: Thanks for pointing this out. We have now added the sample sizes (i.e., number of occurrence records) used to reconstruct the species’ geographic ranges under the different methods in a supplementary table (Table S1) and included a short paragraph in the results section of the main document indicating relevant samples sizes (highest and lowest).

In particular, the reviewer was concerned on the use of three occurrence points/records to characterize species range sizes and their climatic variation. First, we note that only four out of the 103 studied species had their geographic ranges reconstructed using three points, for which we used only the polygon methods (which require a minimum of three records to do so). For SDMs, the lowest sample size was seven records. Although these sample sizes may seem small, they are statistically feasible and our observed range size estimates were extremely consistent across the different methods employed. Accordingly, as we mentioned in our main text, the further evaluation of ecological hypotheses explaining Rapoport’s rule was conducted using only ranges constructed under the alpha-hull method. 

1.3. Overall, the paper is a nice experiment with strong analytical back up. However, it would be really great if you could clarify the basic hypothesis tested here as the title emphasis on testing Rapoport’s rule.

R: We thank the reviewer for their positive feedback and suggestion. To clarify our approach and hypotheses, we have now improved our explanation for each of the four hypotheses tested in the revised main text. Overall, several instances of change in our revised version were aimed at clarifying our two-fold objective of testing for Rapoport’s rule and evaluating its proposed environmental drivers.

Reviewer #2 

2.1. It's great to see authors dive into an in-depth analysis to further validate Rapport's rule, especially focusing on latitudinal variation in range sizes using spiny lizards as a model organism. There's indeed a lack of analytical information regarding latitudinal range size variation in lizards, making this study particularly interesting.

The manuscript appears well-written and offers a robust evaluation of four ecological hypotheses, namely historical climate stability, climate variability, climate extremes, and elevation. It's worth noting that the findings strongly support the climate extremes and elevation hypotheses as the primary drivers of spatial variation in range size, surpassing the other two hypotheses.

R: We truly appreciate the reviewer’s supportive feedback and encouragement.

2.2. Nonetheless, it's essential to acknowledge that the reliance on secondary data from sources like the Global Biodiversity Information Facility and Naturalista comes with inherent limitations and potential biases. This is a crucial aspect to consider in the interpretation of the results.

R: The reviewer is correct in that secondary data such as that from GBIF and Naturalista have limitations and can bring biases. We are fully aware of these issues and have now included an additional paragraph in our revised discussion to explicitly acknowledge these limitations that may affect our conclusions. Please, see lines 593-607 in our revised manuscript. 

“Given that our basic study unit, species’ geographic ranges, was based on open data available from secondary sources such as GBIF and Naturalista, it is important to note the limitations and potential biases of this data. Some limitations of data from this sources includes: 1) presence records may be spatially autocorrelated, which can affect the construction of potential distributions using correlative approaches by, for example, estimating greater climate suitability towards heavily sampled regions [107]; 2) presence records often have identification and georeferencing issues [108], potentially resulting in records outside the known species’ range, thus leading to range size overestimations; and 3) some records may come from suboptimal environmental conditions (e.g., sink populations), potentially affecting the estimated climatic conditions occupied by species and thus biasing the evaluation of environmental hypothesis. We tried to avoid these biases as much as possible by conducting a thorough cleaning process, but some inherent biases could remain present. Still, previous studies have demonstrated the feasibility of accurately estimating both species range sizes and climatic variability/extremes using this type of secondary biodiversity data (e.g., [109,110]), thus supporting the use of this repositories in our study.”

2.3. Regarding the methodology section, while it's good that appropriate statistical models were used, it's important to strike a balance between the statistical details and the ecological context. Enhancing the clarity of the methodology section could make the paper more engaging and accessible to a broader audience, reducing the perception of it being solely a statistical manuscript.

R: We understand the reviewer’s concern and have strived to make our methods section more accessible while linking more tightly with the ecological context in several instances along our revised methods section.

Reviewer #3 

Phylogenetic part needs a major revision. The information provided for the phylogenetic reconstruction and divergence dating is severely inadequate. There are multiple details that need to provided:

R: The reviewer is correct in that some details about the phylogenetic hypotheses used in our study were missing. We now provide some of these details (see below), while explicitly acknowledging that we did not conduct any phylogenetic reconstruction. Instead, as its common practice in a vast swath of large-scale studies such as ours, we leveraged on published information about the phylogenetic relationships within our study group to conduct our analyses. Indeed, our goal was not to reconstruct the particular evolutionary history and specific relationships among species of our group, but to conduct phylogenetically informed analyses of the pattern of interest: Rapoport’s rule and its environmental determinants. More specifically, we wanted to describe if our species-level characteristics (e.g., range size, climate variability within range, etc.) exhibited phylogenetic signal (i.e., sister species were more similar than expected by chance) and we were interested in testing the relationship between a response variable, namely range size, and several predictor variables (latitude, climate variability, etc.) at the species level while accounting for their shared ancestry, thus allowing us to explicitly consider the phylogenetic non-independence among observation units (here species) and avoid type 1 error rates. 

To date, there are two species-level phylogenetic hypotheses that include species from our studied genus: 1) the phylogeny of Leaché et al. (2016) at the family level (Phrynosomatidae) and 2) the phylogeny of Tonini et al. (2016) at the order level (Squamata). The idea of using both of these published phylogenies was to confirm that our results were not dependent on the phylogenetic hypothesis used. Our results were consistent, regardless of the phylogeny employed and despite encompassing two different taxonomic scales. Additionally, in this work, we did not use species divergence times for any analysis. However, we agree with the reviewer that it is necessary to clarify details about the construction of the phylogenies to acknowledge their limitations. Still, we note that such details are could be limited to the available information from the original publications.

3.1. There is no information on how the data was aligned, how many genes were used etc.

R: Below, we provide this information for both of the phylogenies used in our study. 

Leaché et al. 2016: The authors used targeted sequence capture data from 585 nuclear loci, of which 541 were from ultraconserved elements and 44 from protein-coding genes. The data alignment process began with demultiplexing using Casava software (Illumina). Then, low-quality sequences were removed using Trimmomatic software. The resulting clean sequences were assembled for each species using IDBA. For cross-species locus assembly, phyluce software was utilized. Multiple sequence alignment was conducted using the MAFFT algorithm, followed by trimming of long and unequal ends to reduce the presence of missing or incomplete data. 

Tonini et al. 2016: For this phylogeny, authors used a total of 17 genes, 7 mitochondrial and 10 nuclear. These include the following categories: ribosomal RNA (12S/16S), amelogenin (AMEL), brain-derived neurotrophic factor (BDNF), bone morphogenetic protein 2 (BMP2), oocyte maturation factor mos (CMOS), cytochrome c oxidase subunit 1 (COI), cytochrome b (CYTB), NADH subunits 1, 2, and 4 (ND1, ND2, and ND4), neurotrophin 3 (NT3), fosducin (PDC), prolactin receptor (PRLR), G protein-coupled receptor 149 (R35), and recombination activating genes 1 and 2 (RAG1 and RAG2). Sequences for these genes were aligned using the “Translation Align” option in Geneious software, with the MAFFT algorithm under default parameters, ensuring that all sequences were coded and in open reading frame.

3.2. What program was used to generate the trees? BEAST? If yes, mention and cite. If using both nuclear and mitochondrial genes, why not use a coalescent approach

R: Below, we provide this information for both of the phylogenies used in our study. 

Leaché et al. 2016: These authors used only nuclear genes. Initially, they used two approaches: concatenation and coalescence. For the concatenation analysis, an unpartitioned maximum likelihood analysis was performed using RAxML v8.0.2 software. They used the GTRGAMMA model, and branch support was estimated using 1000 bootstrap replicates. For the coalescence approach, instead of using a coalescent approach for nuclear and mitochondrial genes, they opted for SVDquartets, a coalescent-based method for inferring phylogenetic trees using complete sequence data. This approach infers topology among randomly selected species quartets using a coalescent model and then assembles the randomly selected quartets into a species phylogenetic tree using a quartet method. For the final tree, where divergence times were calculated, the authors used the BEAST v1.8. software leveraging the relationships inferred through concatenation methods.

Tonini et al. 2016: These authors employed a series of methods for phylogenetic inference and taxonomic assignment to generate fully-sampled phylogenies for the whole order Squamata, including 9574 species. First, they created a maximum-likelihood (ML) topology using ExaML/RAxML for 5415 species based on a molecular supermatrix with sequence data for 17 genes (7 mitochondrial and 10 nuclear). Then, using MrBayes 3.2 (Ronquist et al. 2012), they dated several subclades under a relaxed-clock model with node-age calibrations. Finally, they included the remaining species (those lacking molecular data) to the ML topology by assigning them randomly within their genus or hi

---

## [Decision Letter · Decision Letter 1]

24 Jun 2024

Exploring the causes underlying the latitudinal variation in range sizes: evidence for Rapoport’s rule in spiny lizards (genus Sceloporus)

PONE-D-23-29199R1

Dear Dr. López Reyes,

We’re pleased to inform you that your manuscript has been judged scientifically suitable for publication and will be formally accepted for publication once it meets all outstanding technical requirements along with few comments provided by the academic reviewer.

The revised version of the MS was once again reviewed by the two reviewers who reviewed the previous version. Both the reviewers have been satisfied with the revision but still I feel that some minor changes are needed in few places. I have highlighted those chances to be made in the attached pdf file. Kindly look into it. Additionally, authors are suggested to thoroughly check sentence formation and grammar. Please include these revisions while submitting the final files required for the publication process.

Kind regards,

Prof. Bhoj K. Acharya

Academic Editor

PLOS ONE

Reviewers' comments:

Reviewer's Responses to Questions

**Comments to the Author**

1. If the authors have adequately addressed your comments raised in a previous round of review and you feel that this manuscript is now acceptable for publication, you may indicate that here to bypass the “Comments to the Author” section, enter your conflict of interest statement in the “Confidential to Editor” section, and submit your "Accept" recommendation.

Reviewer #1: All comments have been addressed

Reviewer #3: All comments have been addressed

2. Is the manuscript technically sound, and do the data support the conclusions?

Reviewer #1: Yes

Reviewer #3: Yes

3. Has the statistical analysis been performed appropriately and rigorously? 

Reviewer #1: Yes

Reviewer #3: Yes

4. Have the authors made all data underlying the findings in their manuscript fully available?

Reviewer #1: No

Reviewer #3: Yes

5. Is the manuscript presented in an intelligible fashion and written in standard English?

Reviewer #1: Yes

Reviewer #3: Yes

6. Review Comments to the Author

Reviewer #1: Authors addressed all questions/ comments from the first review and revised the MS to a good shape. I believe it is good to go. Thank you.

Reviewer #3: The authors have addressed all the review comments to my satisfaction. The authors findings contribute valuable insights to the field and align well with the scope and standards of the journal.

7. PLOS authors have the option to publish the peer review history of their article (what does this mean?). If published, this will include your full peer review and any attached files.

Reviewer #1: No

Reviewer #3: No

---

## [Editor Report · Acceptance letter]

28 Jun 2024

PONE-D-23-29199R1 

PLOS ONE

Dear Dr. López-Reyes, 

I'm pleased to inform you that your manuscript has been deemed suitable for publication in PLOS ONE. Congratulations! Your manuscript is now being handed over to our production team.

Kind regards, 

on behalf of

Prof. Bhoj K. Acharya 

Academic Editor

PLOS ONE